# Current-induced magnetization switching in atom-thick tungsten engineered perpendicular magnetic tunnel junctions with large tunnel magnetoresistance

Mengxing Wang[1], Wenlong Cai[1], Kaihua Cao[1], Jiaqi Zhou[1], Jerzy Wrona[2], Shouzhong Peng[1], Huaiwen Yang[1], Jiaqi Wei[1], Wang Kang[1], Youguang Zhang[1], Jürgen Langer[2], Berthold Ocker[2], Albert Fert[1,3] & Weisheng Zhao[1]

Perpendicular magnetic tunnel junctions based on MgO/CoFeB structures are of particular interest for magnetic random-access memories because of their excellent thermal stability, scaling potential, and power dissipation. However, the major challenge of current-induced switching in the nanopillars with both a large tunnel magnetoresistance ratio and a low junction resistance is still to be met. Here, we report spin transfer torque switching in nano-scale perpendicular magnetic tunnel junctions with a magnetoresistance ratio up to 249% and a resistance area product as low as $7.0\,\Omega\,\mu m^2$, which consists of atom-thick W layers and double MgO/CoFeB interfaces. The efficient resonant tunnelling transmission induced by the atom-thick W layers could contribute to the larger magnetoresistance ratio than conventional structures with Ta layers, in addition to the robustness of W layers against high-temperature diffusion during annealing. The critical switching current density could be lower than $3.0\,MA\,cm^{-2}$ for devices with a 45-nm radius.

[1] Fert Beijing Institute, BDBC, and School of Electronic and Information Engineering, Beihang University, 100191 Beijing, China. [2] Singulus Technologies, 63796 Kahl am Main, Germany. [3] Unité Mixte de Physique, CNRS, Thales, Univ. Paris-Sud, Universit´e Paris-Saclay, 91767 Palaiseau, France. Correspondence and requests for materials should be addressed to W.Z. (email: weisheng.zhao@buaa.edu.cn)

Perpendicular anisotropy-based magnetic tunnel junctions (p-MTJs) have great potential for reducing power dissipation and scaling to feature sizes below 20 nm[1–7], and thus have been extensively studied to develop spin-transfer torque magnetic random access memories (STT-MRAMs) and very-large-scale integrated circuits (VLSIs)[8–13]. In particular, p-MTJs with a MgO/CoFeB/heavy metal (e.g., Ta, Hf) structure have attracted interest for their enhanced perpendicular anisotropy that originates from both MgO/CoFeB and CoFeB/heavy metal interfaces[14–18], bringing a reasonable magnetoresistance ratio (TMR) and STT switching critical current density ($J_C$). Furthermore, p-MTJs with a double MgO/CoFeB interface free layer, i.e., MgO/CoFeB/Ta/CoFeB/MgO, have been shown to possess a considerable thermal stability factor ($\Delta$), and $J_C$ comparable to that of p-MTJs with a single interface[19–22].

However, a critical issue in terms of double MgO/CoFeB interfaces is the incorporation of an additional MgO layer, which makes it even more difficult to reduce the resistance area product (RA) below 10 $\Omega\,\mu m^2$, while maintaining a TMR above 150% (see Supplementary Note 1 and Supplementary Fig. 1)[23–26]. On the other hand, for those typical configurations using Ta layers, the TMR, interfacial perpendicular magnetic anisotropy (PMA), and other magnetic properties degrade rapidly at the 400 °C back end of line (BEOL) temperature[27–31]. Thus, there is a need to understand how to enable nano-fabrication compatibility, as well as to reach a compromise between low write energy and large sense margins.

To address those concerns, W was recently reported to replace Ta as spacer and bridging layers in top-pinned p-MTJ films[32–37]. A TMR of 141% after 400 °C annealing and a $\Delta$ of 61 have been obtained from blank films[34]. These improvements were partially attributed to the suppression of atom diffusion and crystalline structure of the W layer, whereas the essential role of W layers in TMR enhancement has not been clearly revealed. Besides, STT switching behaviour and junction resistance were also not shown in these studies.

In our study, a bottom-pinned p-MTJ stack with atom-thick W layers and double MgO/CoFeB interfaces was patterned into nanopillars to demonstrate STT switching with relatively low $J_C$. In addition to the strong thermal endurance shown in Cs-corrected transmission electron microscopy (TEM) observation, a further increased TMR of 249% and an RA as low as 7 $\Omega\,\mu m^2$ are simultaneously achieved. Furthermore, by using the first-principles calculation, atom-thick W layers are found to induce resonant tunnelling transmission more efficiently than Ta layer, providing a comprehensive explanation of the origin for this large TMR.

## Results

**PMA of p-MTJ films.** The p-MTJ stacks we studied here were composed of, from the substrate side, [Co (0.5)/Pt (0.2)]$_6$/Co (0.6)/Ru (0.8)/Co (0.6)/[Pt (0.2)/Co (0.5)]$_3$/W (0.25)/CoFeB (1.0)/ MgO (0.8)/CoFeB (1.3)/W (0.2)/CoFeB (0.5)/MgO (0.75)/Ta (3.0) (Fig. 1a, numbers in parenthesis denote layer thickness in nm), and were deposited on thermally oxidized Si substrate with a 75 nm Ta/CuN/Ta seed layer by a Singulus magnetron sputtering machine. Ultrathin MgO layers were employed to minimize RA. Those p-MTJ films were then subject to vacuum annealing from 350 to 430 °C for an hour. And the p-MTJ film annealed at 390 °C (see Supplementary Note 2 and Supplementary Fig. 2) was patterned into circular nanopillars with 45–150 nm radius ($r$) using electron beam (e-beam) lithography and Ar ion milling, as shown in Fig. 1b.

We investigated the magnetic characteristics of blank samples using a physical properties measurement system-vibrating sample magnetometer (PPMS-VSM). Figure 1c illustrates the representative M–H hysteresis loops under out-of-plane and in-plane magnetic fields, where the p-MTJ film was annealed at 410 °C. The upper and bottom CoFeB free layers present strong ferromagnetic coupling through a 0.2-nm W spacer layer and switch simultaneously according to the minor loop (inset of

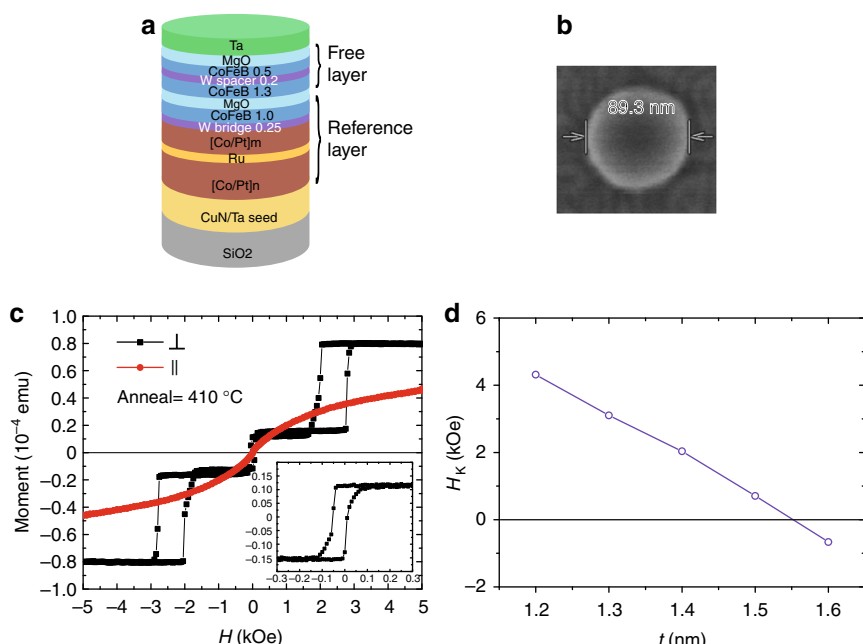

**Fig. 1** Film configuration and magnetic properties. **a** Structure of the p-MTJ stack with MgO/CoFeB/W/CoFeB/MgO free layer and W bridging layer; Co/Pt multilayers are synthetic antiferromagnetic (SAF) layers for bottom pinning. **b** Top view of the p-MTJ pattern ($r = 45$ nm) taken by scanning electron microscope. **c** Out-of-plane ($\perp$) and in-plane ($\parallel$) magnetic fields induced hysteresis loops of the p-MTJ film annealed at 410 °C measured by PPMS-VSM; inset is the minor loop. **d** Dependence of $H_K$ on $t$

Fig. 1c), which is significant to enable STT switching[19]. We also performed a first-principles calculation to study the dependence of the magnetic coupling between the two CoFeB free layers on the thickness of W spacer layer. The p-MTJ stacks reveal the strongest ferromagnetic coupling while using a single-atom W spacer layer, while a transition to weak antiferromagnetic coupling at three-atom W layers[33]. Therefore, the thickness of the 0.2 nm W spacer layer is critical for the STT switching. Moreover, the enhancement of thermal endurance at 410 °C should be related to the lower atom diffusion when using W, instead of Ta, as the spacer and bridging layers[31]. Annealing at higher temperature could improve the crystalline quality of the MgO barrier and the bcc (body-centred cubic) texture of the CoFeB layers[1,38], thus a larger TMR can be expected.

**Thermal stability factor estimation**. To minimize data loss for large memory capacities (e.g., 1 Gb), as well as to meet the industry standard retention time of 10 years, $\Delta > 60$ is required. This factor can be expressed as: $\Delta = \frac{E}{k_B T} = \frac{M_s H_K V}{2 k_B T}$, where $E$ is the energy barrier between two magnetization states, $M_s$ the saturation magnetization, $H_K$ the anisotropy field, $V$ the volume of the free layer, $k_B$ the Boltzmann constant, and $T$ the absolute temperature. To further understand the situation of $\Delta$ in this case, we quantified $H_K$ as a function of CoFeB thickness using the ferromagnetic resonance (FMR) method.

The p-MTJ films with a MgO/CoFeB ($t = 1.2 \sim 1.6$)/W (0.2)/CoFeB (0.5)/MgO free layer were prepared, where $t$ represents the thickness of the bottom CoFeB free layer. For the p-MTJ film with $t = 1.3$ nm (corresponding to the one patterned into nanopillars), $H_K$ is around 3102 Oe, indicating a significant $\Delta \sim 60$ for p-MTJs on 3x-nm technology node (Fig. 1d). Besides, for $t = 1.2$ nm, $H_K$ is as large as 4313 Oe, and a $H_C$ of 70 Oe has been achieved (see Supplementary Note 3 and Supplementary Fig. 3), which is sufficient to overcome the disturbance caused by the read operation. A near-zero shift field suggests an effective reduction of stray field from the Co/Pt SAF reference layer. The p-MTJs have the potential to be further scaled and optimized, specifically for low-power VLSIs and other applications.

**Spin transfer torque in p-MTJs**. Magnetic field and current sweeps were performed at room temperature (295 K) to characterize the STT behaviour in nanopillars. Figure 2a presents the resistance transition of an p-MTJ ($r = 90$ nm) versus magnetic field applied along the out-of-plane direction. The two CoFeB free layers switch as a single layer, and the resistance states are bistable: positive field leads to parallel (P) to anti-parallel (AP) perpendicular magnetization switching, while negative field causes the reverse operation. A TMR as large as 249% is achieved.

Meanwhile, an RA as low as 7 $\Omega\,\mu m^2$ is obtained, which is 40% lower than the typical value of double barrier p-MTJs[19]. Because the RA maintains almost constant with regard to different junction areas, current shunting caused by sidewall redeposition can be excluded. Figure 2b shows STT switching and its detection by resistance change along with DC current sweep. The $J_C$ measured here is +6.0/−5.4 MA cm⁻², which is comparable to that of p-MTJs using Ta spacer and bridging layers; in particular, p-MTJs with $r = 45$ nm show an average absolute $J_C$ around 2.8 MA cm⁻² (see Supplementary Note 4 and Supplementary Fig. 4).

We also characterized STT switching using pulse current with various durations $\tau_P$ at room temperature (295 K). The $J_C$ measured from the p-MTJ (TMR = 237% in Fig. 3a) with $r = 75$ nm is +6.9/−6.5 MA cm⁻² at $\tau_P = 100\,\mu s$ (Fig. 3b). Additionally, we characterized $J_C$ as a function of $\tau_P/\tau_0$, where $\tau_0 = 1$ ns is the characteristic attempt time. As plotted in Fig. 3c, the intrinsic critical current density $J_{C0}$ is fitted as 7.8 MA cm⁻². Because the exchange coupling between two magnetic layers has a dependence on the temperature[39], this measurement was also conducted at 35 K (see Supplementary Note 5 and Supplementary Fig. 5), which can also eliminate the impact of thermally activated transition.

**First-principles calculation of TMR**. We theoretically explain this high TMR by the first-principles calculation, which combines the Keldysh nonequilibrium Green's function with the density functional theory (NEGF-DFT)[40]. This technique has been used in our preliminary TMR calculation[41]. Here, atomic structures were built according to our experimental p-MTJ configuration, i.e., Ta (001)/CoFe (001)/X/CoFe (001)/MgO (001)/CoFe (001)/Ta (001), where the X represents W or Ta spacer layers for comparison (see Supplementary Note 6 and Supplementary Fig. 6).

The computed TMRs for the MTJ stacks with single-atom W or Ta spacer layer are 245% and 90%, respectively, which are consistent with our experiments and the previous results based on Ta layers[19–21]. Spin-resolved conductance was obtained by using Landauer–Büttiker formula: $G_\sigma = \frac{e^2}{h} \sum_{k_\parallel} T_\sigma(k_\parallel, E_F)$, where the $\sum_{k_\parallel} T_\sigma(k_\parallel, E_F)$ is the transmission coefficient at the Fermi level $E_F$ with spin $\sigma$ and transverse Bloch wave vector $k_\parallel = (k_x, k_y)$; $e$ is the electron charge and $h$ is the Planck constant. Here, we plotted the transmission spectrums with log scale in the Brillouin zone as shown in Fig. 4 for analysis, and the colour bar shows the transmission probability from low (blue) to high (red). It can be seen that for the majority spin in the P state (Fig. 4a, e), a broad peak centred at $k_\parallel = (0, 0)$ appears due to the slow decay of $\Delta_1$ state. For the minority spin in P state (Fig. 4b, f), sharp peaks called hot spots appear at edges (shown in the red circle). This is

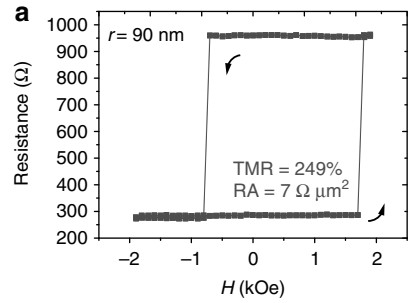
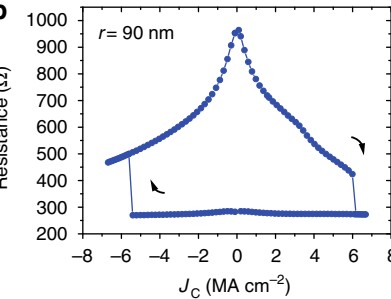

**Fig. 2** Magnetoresistance and STT measurements for p-MTJ ($r = 90$ nm) at room temperature. **a** Magnetoresistance as a function of out-of-plane magnetic field and **b** STT switching measured by DC current sweep. Arrows show the perpendicular magnetization transitions from AP to P states or the opposite situation

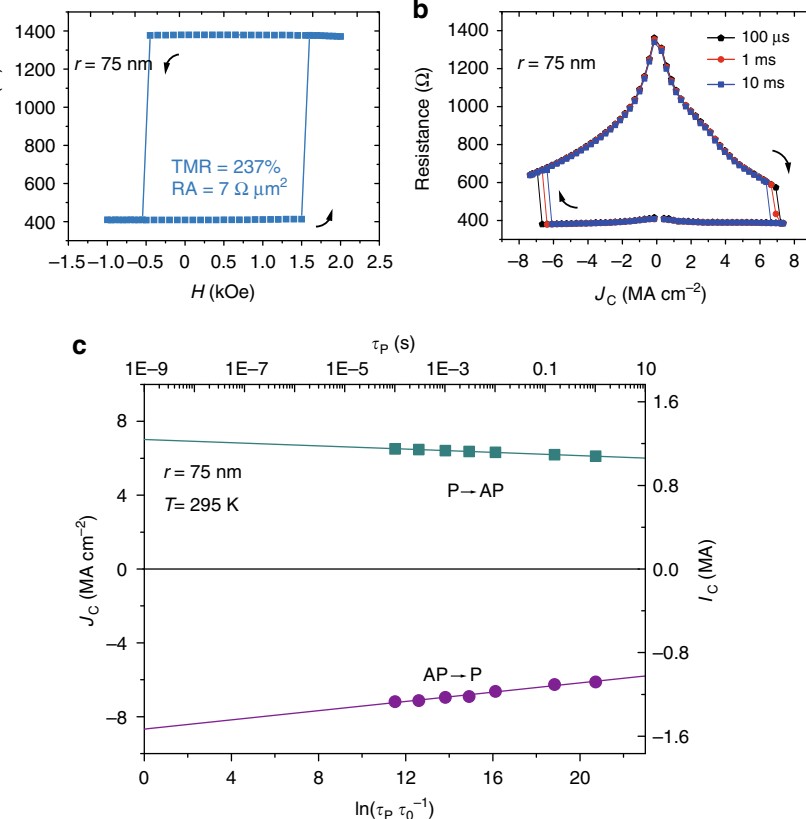

**Fig. 3** Magnetoresistance and STT measurements for p-MTJ ($r = 75$ nm) after optimization at room temperature. **a** Magnetoresistance as a function of out-of-plane magnetic field. **b** STT switching measured with pulse current at various $\tau_P$. **c** $J_C$ as a function of $\ln(\tau_P/\tau_0)$. Arrows show the perpendicular magnetization transitions from AP to P states or the opposite situation

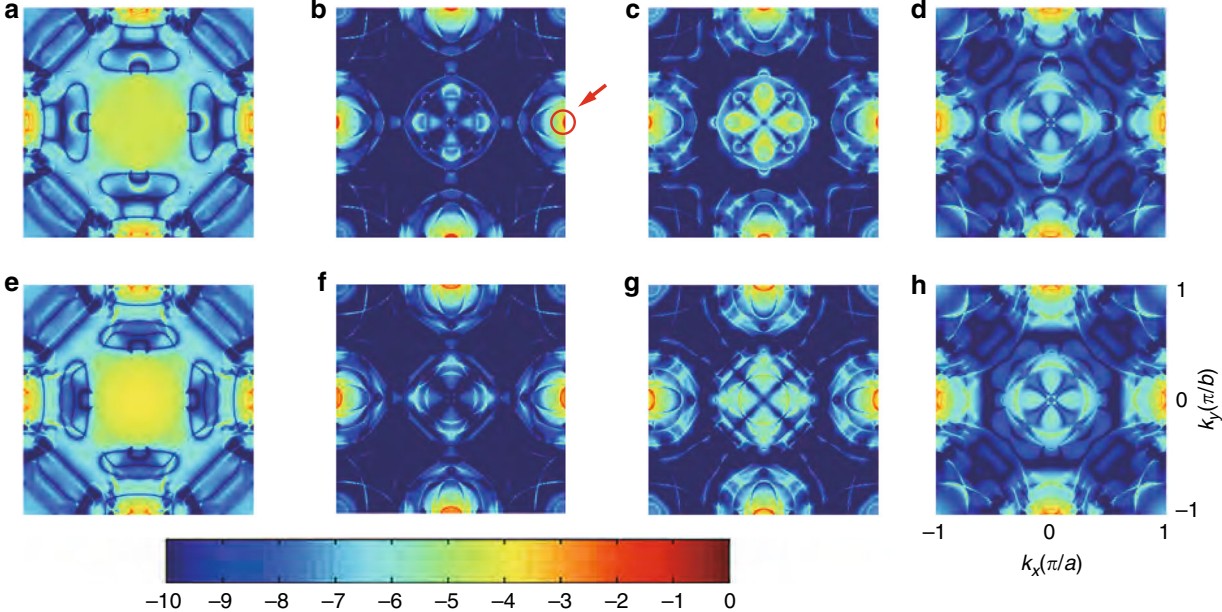

**Fig. 4** Spin- and $\mathbf{k}_{\parallel}$- resolved transmission coefficients. Transmission spectrums for p-MTJ stacks with **a–d** W, and **e–h** Ta spacer layers. **a**, **e** present the majority-to-majority conditions, and **b**, **f** the minority-to-minority conditions in P state; **c**, **g** present the majority-to-minority conditions, and **d**, **h** the minority-to-majority conditions in AP state

caused by the resonant tunnelling transmission, which occurs when the localized interface states on the two CoFe/MgO interfaces align in energy[42]. Whereas in the AP state (Fig. 4c, d,

g, h), the transmission coefficients of the hot spots are relatively lower.

As the resonant tunnelling transmission is sensitive to the bias voltage ($V_b$), we further calculated the dependence on $V_b$ of TMR.

For the p-MTJ with W spacer layer, the TMR dramatically drops with increasing $V_b$: when $V_b = 10$ mV, the TMR is 173%; and when $V_b = 50$ mV, the TMR has decreased to 107%. On the contrary, for the p-MTJ with Ta spacer layer, the TMR remains 88% at $V_b = 50$ mV. This intense TMR decay is consistent with our experimental results (e.g., Fig. 2b), while the p-MTJ using Ta insertion presents less dependency[22]. Accordingly, we conclude that the resonant tunnelling transmission with higher transmission coefficient could contribute to the large TMR for the p-MTJ stack with W spacer layer.

To establish a more distinct physical picture, we present how the scattering state, which is the absolute square of the tunnelling electron wavefunction, changes at CoFe/MgO interfaces (see Supplementary Note 7 and Supplementary Fig. 7). And the transmission probability is proportional to the density of scattering states. In the case of atom-thick W spacer layer, a higher density of scattering states is obtained at the region around the $k_{||} = (0, 1)$ point, leading to a larger conductance in the P state.

**Crystallization and atom distribution study.** Spherical aberration corrected TEM (Cs-corrected TEM) and atomic-resolution electron energy-loss spectroscopy (EELS) were applied to study the structural properties of the p-MTJ stacks. Figure 5a shows a Cs-corrected TEM image of the p-MTJ stack annealed at 390 °C, which verifies the excellent crystalline quality of the MgO barrier, though the atom-thick W spacer and bridging layers are too thin to be captured. Figure 5b maps the EELS intensities of W, B, and Mg after 410 °C annealing, and no distinct change of W distribution is shown compared with the nominal locations of W spacer and bridging layers, especially at increasing temperatures (see Supplementary Note 8 and Supplementary Fig. 8). Because W is a heavy metal element, we established an energy-dispersive X-ray spectroscopy EDS test as a further confirmation. As mapped in Fig. 5c, no signal of W was detected within the MgO barrier or at the CoFeB/MgO interfaces, supporting the point that W is robust against high-temperature diffusion. Besides, the peaks of B and W are quite close in the EELS profiles, indicating a large amount of B existing in the W spacer and bridging layers. Thus, the W layers not only provide a typical bcc template for the texture of adjacent CoFeB layers, but also absorb B atoms during annealing to create robust interfacial PMA. Both the crystalline structure and atom distribution contribute to the interfacial PMA, TMR, and thermal endurance.

## Discussion

Overall, the TMR enhancement we have achieved mainly originates from two factors: from the viewpoint of mechanism, the efficient resonant tunnelling induced by the atom-thick W layer could contribute to a larger TMR than the conventional p-MTJs with Ta layers; on the other hand, the W layer is robust against high-temperature diffusion, resulting in better crystallinity of MgO barrier and higher TMR.

In addition, the single-atom W layers we used also benefit the p-MTJ nano-pillars from the following aspects. First, we have experimentally and theoretically revealed that the largest TMR can be achieved by using single-atom W layers under this situation. The p-MTJ films with 0.2 nm W spacer layer possess TMRs about 20% higher than that of samples using 0.3 nm (see Supplementary Fig. 2a). This corresponds to the tendency obtained from the first-principles calculation: the calculated TMR decreases from 245% to 171% while increasing the thickness of W from a single-atom layer to three atoms (see Supplementary Note 9 and Supplementary Table 1).

Second, the $J_C$ we obtained is comparable to that for p-MTJs configured with Ta layers, and does not scale with the enhanced $\Delta$, which can be ascribed to the lower damping constant ($\alpha$). The $\alpha$ of conventional p-MTJs using Ta layers is sub-optimal for its strong spin–orbit coupling and atom diffusion during high-temperature annealing. Our earlier experiments have proved that $\alpha$ is material dependent, thus could be reduced by replacing Ta with W as the spacer layer[43]. Further, the thinner the W spacer layer is, the lower the $\alpha$ becomes, because W diffusion is weakened with decreasing thickness[44].

Third, the 0.2 nm W spacer layer enables the two CoFeB free layers to switch as a single layer in STT measurement by inducing a strong ferromagnetic coupling; and the 0.25 nm W bridging layer can ensure the stable ferromagnetic coupling between CoFeB reference layer and SAF layers. It has been widely investigated that the exchange coupling between two ferromagnetic layers separated by a nonmagnetic interlayer exhibits oscillatory behaviour due to the RKKY (Ruderman–Kittel–Kasuya–Yosida) iteration[39,45–47]. Moreover, as we calculated, as the nonmagnetic interlayer becomes thicker, the ferromagnetic coupling weakens monotonically and converts to antiferromagnetic coupling at three-atom W layers. Therefore, the atom-thick W spacer layer is critical for the switching reliability of our device regarding STT performance. However, our p-MTJ stack is a complex system involving stray

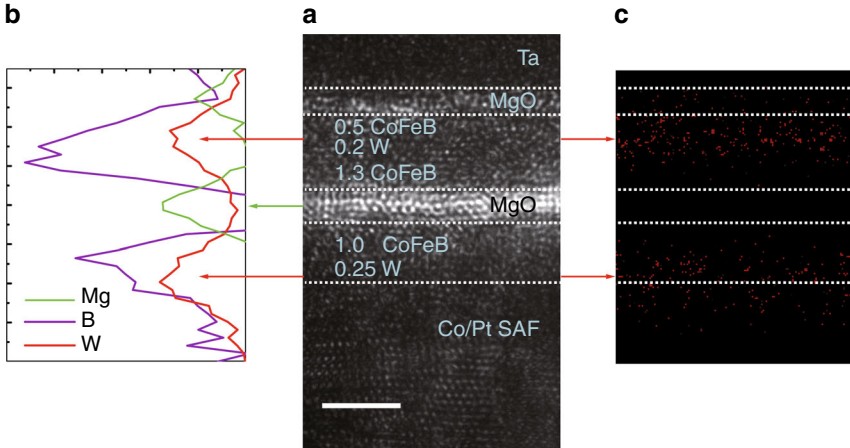

**Fig. 5** Cs-corrected TEM and EELS results. **a** Cs-corrected TEM image that profiles the crystallization. The p-MTJ stack was annealed at 390 °C. The scale bar indicates 2 nm. **b** EELS intensities of Mg, B, and W. Arrows show the positions of the same layer in the two figures. **c** EDS mapping of the p-MTJ stack, where W is in red

field and other factors, thus more specific experiment should be carried out to prove this tendency.

To conclude, we demonstrated for the first time current-induced magnetization switching in p-MTJs with atom-thick W spacer and bridging layers, which present a large TMR of 249% and an RA as low as $7\,\Omega\,\mu m^2$. In particular, the experimental investigations and theoretical analyses provide an insight into the role of atom-thick W layers in determining TMR. We believe that this work provides a critical path to the research and development of new generation STT-MRAM (see Supplementary Note 1 and Supplementary Fig. 1).

## Methods

**Film growth.** The p-MTJ stacks in our work were mainly grown with linear dynamic deposition technology by a Singulus TIMARIS 200 mm magnetron sputtering machine at a base pressure of $3.75 \times 10^{-9}$ Torr. The substrates were thermally oxidized Si with a 75-nm Ta/CuN/Ta seed layer polished by chemical mechanical planarization. MgO deposition was performed by RF sputtering. The base pressure for vacuum annealing oven is around $3.75 \times 10^{-10}$ Torr.

**Device fabrication.** To observe the STT effect, nanopillars were defined by e-beam lithography in the centre of 4-μm-wide bottom electrodes followed by Ar ion milling. Then they were fully covered with $SiO_2$ for insulation. After the lift-off procedure, via holes were made over the bottom electrodes. Both the bottom electrodes and p-MTJs were then connected to 90 nm Ti/Au electrodes to allow electrical contact for measurement using e-beam evaporation.

**Magnetic and electrical measurement.** The blank stacks were studied with PPMS-VSM and FMR systems. The Cs-corrected TEM was performed by JEM ARM 200F. The PPMS-VSM used is the Quantum Design VersaLab. The setup for current-induced p-MTJ switching using the four-probe method consists of a Lake Shore CRX-VF cryogenic probe station, a Keithley 6221 current source, and a 2182 nanovolt metre.

**Data availability.** All data generated or analyzed during this study are included in this published article (and its Supplementary Information).

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

## Acknowledgements

The authors gratefully acknowledge the National Natural Science Foundation of China (Grant No. 61571023, 61627813, 61501013), International Collaboration Project B16001 and 2015DFE12880, and Beijing Municipal of Science and Technology (Grant No. D15110300320000) for their financial support of this work. The authors also would like to thank Lin Gu and Qinghua Zhang from Beijing National Laboratory for Condensed Matter Physics, Institute of Physics, Chinese Academy of Sciences, for the technical support of Cs-corrected TEM.

## Author contribution

W.Z. conceived and supervised the project. M.W. fabricated the devices, initiated the measurements. M.W., and W.Z. wrote the manuscript. W.C. performed the measurements. K.C., and J.W. optimized the e-beam lithography flow. J.Z., and S.P. carried out the first-principles calculation. H.Y., W.K., Y.Z., and A.F. helped analyse the data. J.W., J.L., and B.O. developed, grew and optimized the films. All authors discussed the results and implications.

## Additional information

**Competing financial interests:** The authors declare no competing financial interests.

