## [Peer Review File · Nature Communications]

Reviewers' comments:

Reviewer #1 (Remarks to the Author):

The authors report large tunnel magnetoresistance (TMR) ratios and a moderate critical current density of spin-transfer-torque (STT) magnetization switching for perpendicularly magnetized magnetic tunnel junctions (MTJs) with W atomic layers. I think the MTJ performance obtained in the present study is really excellent. The number of 249% in TMR ratios is remarkable for the MRAM application, and the demonstration of STT switching is so attractive from an application point of view. In addition, the data and the analysis of STT switching seem to be properly obtained and performed, respectively. The first principles calculations for a model structure agree with the experimental observations, suggesting that the W atom plays an important role on the electronic states that mainly contribute to the electron transport. So, I feel that the present results obtained are really valuable, particularly in the field of spintronic application. At the same time, I suppose that it is unclear whether the theoretical calculations physically explain the observed TMR ratios since the CoFeB layers contain a lot of boron atoms and are in an amorphous or amorphous-like state, in contrast to the defect-free lattice assumed in the calculation model. If the excellent results obtained did not come from the modification of electron transport by adding the W atoms but was due to the ingenious optimization process for the whole MTJ structures, the scientific value of the paper may be limited. A couple of specific comments are as follows:

- (1) More systematic calculations would strengthen the authors' discussion on the mechanism of the enhanced TMR. For example, a few other structural models including small defects can be used to investigate the effect of the lattice distortion and/or defects. If the bias voltage dependence of dI/dV and/or TMR is compared between the calculations and the experiments, better understanding is likely to be obtained.
- (2) If the W atomic layer would remain even after annealing processes, the state-of-the-art TEM technique could visualize an image of the W layer, since W atoms show a strong contrast due to their high core electron densities.

Reviewer #2 (Remarks to the Author):

This paper reviewed the effect of the tungsten spacer layer (authors defined this spacer layer as capping layer which is wrong definition) on the TMR enhancement, compared to the (tantalum)

spacer layer. This effect has been already reported at several papers. Author did not contribute a novel p-MTJ structure or new scientific discovery.

1. In abstract, author mentioned: Besides, the robustness of W layers against high temperature diffusion avoids TMR degradation during annealing. (line 15~17) However, I could not find the experimental result of this sentence. How can this be mentioned in the abstract without any experimental data to back this sentence.

2. In figure 1(a), this p-MTJ structure is not a new one. How can future audiences obtain a new scientific discovery when they read your paper.

3. In figure 1(c), author mentioned the TMR ratio peaked at 410°C. The author should explain this result more thoroughly and show the scientific evidence.

4. In figure 2, author mentioned the TMR ratio could be enhanced by applying an additional perpendicular magnetic field of 2kOe prior to the TMR measurement. This is not realistic method for operating the p-MTJ spin-valve. Also, does the author have any experimental evidence of the spin angle distribution in the free layer between with and without additional perpendicular magnetic field of 2kOe.

5. In figure 3 and supplementary figure 4, the experimental data are general information that activation energy between two different states is inversely proportional to the temperature. What did the author like to review scientifically in figure 3 and supplementary figure 4.

6. In figure 4, authors demonstrated the electron tunneling ability of the minority-to-minority transition, which showed that the tungsten spacer showed higher electron tunneling ability of minority-to-minority transition than that of tantalum. However, it was more associated with the f.c.c crystallinity of the MgO tunneling barrier influenced by the spacer material such as tungsten or tantalum; i.e., the tungsten spacer showed much better f.c.c. crystallinity tunneling barrier than the tantalum spacer. [D.Y. Lee, et al. Scientific Reports 6, 38125 (2016), Lee, S. E., Shim, T. H., & Park, J. G., NPG Asia Mater. 9, e324 (2016)]

7. In figure 5, the quality of the cross-sectional TEM image of p-MTJ spin-valve is very poor. Thus, this TEM image could not discuss about the f.c.c. crystallinity of the MgO tunneling barrier.

In conclusion, authors did not show any new scientific discovery or any advanced technology in p-MTJ spin-valves.

Reviewer #3 (Remarks to the Author):

In this paper, the authors studied the STT of a p-MTJ with an atomic layer of Tungsten experimentally. It was claimed that the device has large TMR of 249% and at the same time a resistance area product lower than 7.5 ohm micron². First principles study has identified physical origin of high TMR. This is a high quality work that is suitable for nature comm. However, I have the following concerns that must be clarified.

The punch point of this paper is the achievement of large TMR of 249% and low resistance area product (denoted as RA) lower than 7.5. In Fig.2, results of two devices (the same radius) have been given in Fig.2a and Fig.2c. In Fig.2a, the TMR is 161% while RA is about 7 (see first paragraph on page 5). In Fig.2c, the TMR is 249% while RA is not specified.

Here is my question, looking at Fig.2, the resistance are around $R=720$ and 950 for Fig.2a and Fig.2c, respectively. Since two devices have the same radius, the RA of Fig.2c is clearly larger than 9 or even close to 10. I would like to know how RA was calculated for Fig.2a, $RA=720 \cdot \pi \cdot r^2$?

To conclude, the statement “TMR of 249% and an RA as low as $7 \Omega \cdot \mu\text{m}^2$ are simultaneously achieved” is inaccurate. This makes this work not as significant (see Fig.1 of supplementary material).

Here are other questions:

- (1). In the measurement what is the bias applied?
- (2). Caption for Fig.2c and Fig.2d are missing.
- (3). The resonant tunneling was identified as the origin of high TMR. Since resonant tunneling is vulnerable to bias and impurity, this issue has to be addressed.
- (4). Fig.2c, in the inset it says TMR=248% while TMR=249% in the text.
- (5). Since the presence of W layer is detrimental to resonant tunneling (make the system asymmetric), what is the role play by W, to help stabilize the system? Or to help the system to crystalize?

Reviewer #1 (Remarks to the Author):

The authors report large tunnel magnetoresistance (TMR) ratios and a moderate critical current density of spin-transfer-torque (STT) magnetization switching for perpendicularly magnetized magnetic tunnel junctions (MTJs) with W atomic layers. I think the MTJ performance obtained in the present study is really excellent. The number of 249% in TMR ratios is remarkable for the MRAM application, and the demonstration of STT switching is so attractive from an application point of view. In addition, the data and the analysis of STT switching seem to be properly obtained and performed, respectively. The first principles calculations for a model structure agree with the experimental observations, suggesting that the W atom plays an important role on the electronic states that mainly contribute to the electron transport. So, I feel that the present results obtained are really valuable, particularly in the field of spintronic application.

Response: Thank you very much for the positive comments. We appreciate your approval of our work regarding STT switching in atom-thick engineered p-MTJs with large TMR and low RA. We would like to thank you for evaluating our work as remarkable and attractive. We also thank you for the instructive suggestions, which helped us enhance the quality of the manuscript. We hope our work is presented more clearly after the revision.

At the same time, I suppose that it is unclear whether the theoretical calculations physically explain the observed TMR ratios since the CoFeB layers contain a lot of boron atoms and are in an amorphous or amorphous-like state, in contrast to the defect-free lattice assumed in the calculation model. If the excellent results obtained did not come from the modification of electron transport by adding the W atoms but was due to the ingenious optimization process for the whole MTJ structures, the scientific value of the paper may be limited.

Response: Thank you. As seen from the EELS results in Fig. 5b and Supplementary Fig. 8, the peaks of B and W were quite close, indicating that B immigrated into W insertions during annealing and CoFe with bcc texture consequently formed. This phenomenon was also observed by other research groups.^{1,2} Thus, using CoFe instead of CoFeB for the MgO/CoFeB/capping layer atomic structure was quite common in the first-principles calculations.^{3,4} Moreover, the Cs-corrected TEM (Fig. 5a) in the revised manuscript showed that the crystal lattice of the CoFe(B) layers may exist. Though the calculation cannot consistently model the situation of the p-MTJ films, the revealed tendency and mechanism can still qualitatively explain the experimental results. The calculations actually focused more on the comparison between p-MTJs

using atom-thick Ta or W layers instead of that between the experimental and calculated results.

As the calculated transmission spectra in Figs. 4b and f showed, sharp peaks called hot spots, appeared at the edges for the minority spin in the P state, which was caused by the resonant tunnelling transmission. Referring to the colour bar, we can tell that the transmission probability at the hot spots was higher in the case of the W insertion than that in the Ta case. Accordingly, we conclude that the resonant tunnelling transmission with higher transmission coefficient had more benefits for the conductance in the P state, thereby leading to a larger TMR for the p-MTJ stack with the W insertion.

A couple of specific comments are as follows:

(1) More systematic calculations would strengthen the authors' discussion on the mechanism of the enhanced TMR. For example, a few other structural models including small defects can be used to investigate the effect of the lattice distortion and/or defects. If the bias voltage dependence of dI/dV and/or TMR is compared between the calculations and the experiments, better understanding is likely to be obtained.

Response: Thank you very much for this suggestion, which helped us further confirm the contribution of resonant tunnelling transmission for the high TMR of the W spacer layers. As you have proposed, we further calculated the dependence on the bias voltage (V_b) of TMR using the atomic structures we built in the main text. The resonant tunnelling transmission was sensitive to the V_b . Hence, for the p-MTJ with the W insertion, the TMR dramatically dropped with the increasing V_b : the TMR was 173% when $V_b = 10$ mV and decreased to 107% when $V_b = 50$ mV. On the contrary, the TMR for the p-MTJ with the Ta insertion remained to be 88% at $V_b = 50$ mV. This intense TMR decay associated with W insertion was also observed in Figs. 2b, d and Fig. 3b in the manuscript. The p-MTJ using the Ta insertion presented less dependency.⁵ Therefore, we think that the W insertion-induced resonant tunnelling transmission indeed contributed to the TMR enhancement.

(2) If the W atomic layer would remain even after annealing processes, the state-of-the-art TEM technique could visualize an image of the W layer, since W atoms show a strong contrast due to their high core electron densities.

Response: We apologise for the limitation of our HRTEM image. We have performed a Cs-corrected TEM in the revised manuscript to improve its quality, of which the resolution was enhanced to 0.1 nm. Fig. 5a shows that the crystal lattice of the MgO

barrier has been well profiled, while the 0.2 nm and 0.25 nm W insertions still cannot be visualised. The situations were quite similar for the p-MTJ films using even thicker 0.55 nm or 0.4 nm W layers.^{6,7} While as a supplement, the existence of the atom-thick W layers can be read from the EELS profiles in Fig. 5b, where the peaks of the atom-thick W spacer and the bridging layers are distinctly displayed.

References

1. An, G. G. *et al.* Highly stable perpendicular magnetic anisotropies of CoFeB/MgO frames employing W buffer and capping layers. *Acta Mater.* **87**, 259-265 (2015).
2. Wang, Z. *et al.* Atomic-Scale Structure and Local Chemistry of CoFeB-MgO Magnetic Tunnel Junctions. *Nano Lett.* **16**, 1530-1536 (2016).
3. Khoo, K. H. *et al.* First-principles study of perpendicular magnetic anisotropy in CoFe/MgO and CoFe/Mg₃B₂O₆ interfaces. *Phys. Rev. B* **87**, 174403 (2013).
4. Zhang, J., Franz, C., Czerner, M., & Heiliger, C. Perpendicular magnetic anisotropy in CoFe/MgO/CoFe magnetic tunnel junctions by first-principles calculations. *Phys. Rev. B* **90**, 184409 (2014).
5. Devolder, T., Le Goff, A., & Nikitin, V. Size dependence of nanosecond-scale spin-torque switching in perpendicularly magnetized tunnel junctions. *Phys. Rev. B* **93**, 224432 (2016).
6. Kim, J. H. *et al.* Ultrathin W space layer-enabled thermal stability enhancement in a perpendicular MgO/CoFeB/W/CoFeB/MgO recording frame. *Sci. Rep.* **5**, (2015).
7. Lee, D. Y., Hong, S. H., Lee, S. E., & Park, J. G. Dependency of Tunnelling-Magnetoresistance Ratio on Nanoscale Spacer Thickness and Material for Double MgO Based Perpendicular-Magnetic-Tunnelling-Junction. *Sci. Rep.* **6**, (2016).

Reviewer #2 (Remarks to the Author):

This paper reviewed the effect of the tungsten spacer layer (authors defined this spacer layer as capping layer which is wrong definition) on the TMR enhancement, compared to the (tantalum) spacer layer. This effect has been already reported at several papers. Author did not contribute a novel p-MTJ structure or new scientific discovery.

Response: Thank you so much for the comments. Before answering the first question, we would like to mention the two major progresses shown in this work, and hope you give a second thought:

1. For the first time, we achieved a TMR of up to 249% in a perpendicular magnetic tunnel junction (p-MTJ) nanopillar switched by spin transfer torque (STT). No any other prior work reported on STT in MTJ with a TMR higher than 180%.¹
2. We identified that the contribution of resonant tunnelling transmission is one of the major reasons for the obtained high TMR. Both the experiments and the theoretical calculation confirmed this point.

In this revised manuscript, we have renamed the W insertion as “spacer layer” following your suggestion. However, we think that it is still an open question to define this layer. In the last years, numerous experimental works have shown that the hybridization of one atom heavy metal layer with one ferromagnetic metal layer can strongly affect the magnetic properties.²⁻⁴ We have also theoretically verified that the thin heavy metal layer has a great influence on interfacial PMA and TMR, so does the W insertion in this case.⁵⁻⁷ Thus, the W insertion in the CoFeB/W/CoFeB structure plays a role much more than as a spacer layer. While to avoid misunderstanding, we have changed the capping layer into “spacer layer” in the revised manuscript.

1. In abstract, author mentioned: Besides, the robustness of W layers against high temperature diffusion avoids TMR degradation during annealing. (line 15~17) However, I could not find the experimental result of this sentence. How can this be mentioned in the abstract without any experimental data to back this sentence.

Response: We apologise for this unclear statement in the manuscript. This conclusion can be drawn from the EELS profiles in Fig. 5b and Supplementary Fig. 8. First, as seen in Fig. 5b, the distribution of the W peaks was consistent with the locations of the W spacer and bridging layers in the design structure. Second, the locations of the W peaks remained almost the same when increasing the annealing temperature from 370 °C to 410 °C (Supplementary Fig. 8). These results indicated that the diffusion of the W atoms at a higher temperature was suppressed, which should benefit both

interfacial PMA and TMR. We have reconfigured the figures for a better understanding.

2. In figure 1(a), this p-MTJ structure is not a new one. How can future audiences obtain a new scientific discovery when they read your paper?

Response: From the structure point of view, our p-MTJ stack possessed several major changes compared to those previously published W layer-based structures.

1. The thicknesses of the two MgO layers are only 0.8 nm and 0.75 nm, which is quite significant to achieve the lower RA in this case; and this is also critical for STT mechanism in p-MTJ.

2. We used various thicknesses for the lower and upper CoFeB layers in the free layer, which broke the symmetry compared to the previous structures. The 1.3 nm CoFeB layer was designed to favour STT switching by reducing J_C , and it is also a critical compromise to realize high TMR and maintain interfacial PMA at the same time. The 0.5 nm CoFeB was adjusted to enhance PMA and thermal stability. These two CoFeB layers were ferromagnetically coupled together.

3. The thickness of the W spacer layer is only 0.2 nm in this case. We have experimentally and theoretically revealed that the largest TMR can be achieved by using single-atom W layers. The p-MTJ films with 0.2 nm W spacer layer possess TMRs about 20% higher than that of samples using 0.3 nm (see Supplementary Fig. 2a). And the calculated TMR decreases from 245% to 171% when increasing the thickness of W from single-atom layer to three. Our earlier experiments have proved that the damping constant α is material dependent, thus could be reduced by replacing Ta with W as the spacer layer.⁸ Further, the thinner the W spacer layer is, the lower the α becomes, because W diffusion is weakened with decreasing thickness.⁹ Thus, the J_C we obtained was comparable to that for the p-MTJs configured with the Ta layers and did not scale with the enhanced Δ . Additionally, the 0.2 nm W spacer layers may strengthen the ferromagnetic coupling between the upper and bottom CoFeB free layers, because as the nonmagnetic interlayer becomes thicker, the ferromagnetic coupling weakens monotonically and converts to antiferromagnetic coupling.¹⁰

From the demonstration point of view, we presented, for the first time, the STT switching in the W layer engineered p-MTJ nanopillars. The previous work went into TMR and thermal stability, which were evaluated from blank films. Never has a nanopillar been fabricated for the STT measurement. In our work, we successfully patterned p-MTJ nano-pillars and characterised and analysed the situation of STT switching. We believe that this work is absolutely original and meaningful, especially when considering the final application in STT-MRAM.

3. In figure 1(c), author mentioned the TMR ratio peaked at 410°C. The author should explain this result more thoroughly and show the scientific evidence.

Response: We apologise for this unclear expression. The word “maximum” that we used may be vague. We have rewritten this phrase as “it increased to a higher value of 179%”. In the revised manuscript, we have also analysed the dependence of the TMR on the annealing temperature. The TMR enhancement along with the increasing annealing temperature (e.g., from 350 °C to 410 °C in this case) has been widely reported to be caused by the improved crystalline quality of the MgO barrier, as well as the bcc texture of the CoFeB layers.^{11,12} Drastic atom diffusion happens with the further rising temperature (e.g. from 410 °C to 430 °C in this case), including Co, Fe, W and the Ta/CuN used for seed and capping layers in our stack.^{4,13,14} This diffusion leads to the formation of a magnetic dead layer and a weaker bonding between the Fe-3d and O-2p orbitals.¹⁵ Thus, both TMR and interfacial PMA degrade above 410 °C annealing.

4. In figure 2, author mentioned the TMR ratio could be enhanced by applying an additional perpendicular magnetic field of 2 kOe prior to the TMR measurement. This is not realistic method for operating the p-MTJ spin-valve. Also, does the author have any experimental evidence of the spin angle distribution in the free layer between with and without additional perpendicular magnetic field of 2 kOe.

Response: Thank you. We have reorganised this paragraph for better understanding. We would like to clarify that this enhancement is permanent after the optimisation. Hence, the method can be potentially applied to the p-MTJ operation. Figs. 2a and c show that the H_C was enlarged from 470 Oe to 1200 Oe after optimisation, suggesting the strengthened perpendicular magnetic moment. Meanwhile, the J_C also increased (Figs. 2b and d) when θ tended to $0/\pi$, which can be explained by Eqs. (3) to (5) in Ref. 16. All data in Fig. 2 were measured from the same p-MTJ.

5. In figure 3 and supplementary figure 4, the experimental data are general information that activation energy between two different states is inversely proportional to the temperature. What did the author like to review scientifically in figure 3 and supplementary figure 4?

Response: Thank you. The intrinsic threshold current density J_{C0} can be estimated by performing the magnetoresistance-pulse current measurement in Fig. 3. Furthermore, we also conducted this measurement at 35 K (Supplementary Fig. 4) because the low temperature affected the exchange coupling between the two CoFeB layers in the free layer. We can conclude that the J_C in this case was comparable to that of the

conventional p-MTJ nanopillars using Ta layers. The test was a general method for the STT study, but the research object and the revealed information were quite new, filling the gap regarding the STT behaviour in the p-MTJ nanopillars using atom-thick W spacer and bridging layers.

6. In figure 4, authors demonstrated the electron tunnelling ability of the minority-to-minority transition, which showed that the tungsten spacer showed higher electron tunnelling ability of minority-to-minority transition than that of tantalum. However, it was more associated with the f.c.c crystallinity of the MgO tunnelling barrier influenced by the spacer material such as tungsten or tantalum; i.e., the tungsten spacer showed much better f.c.c. crystallinity tunnelling barrier than the tantalum spacer. [D.Y. Lee, et al. Scientific Reports 6, 38125 (2016), Lee, S. E., Shim, T. H., & Park, J. G., NPG Asia Mater. 9, e324 (2016)]

Response: Thank you. Yes, these two references provided strong evidence and proved the positive effect on the crystallinity of the MgO barrier from the W spacer layer. We have updated our references following your comment. We have also theoretically verified that the efficient resonant tunnelling transmission induced by the W spacer layer contributes to the TMR enhancement. As mentioned in the HRTEM image, we are not denying the influence of the W layer on the crystallinity of the adjacent layers. It is possible that the well-textured MgO/CoFeB interfaces benefit the resonant tunnelling transmission, resulting in a larger TMR in our p-MTJ stack. Thus, more investigations may be needed to determine which factor dominates in this case.

7. In figure 5, the quality of the cross-sectional TEM image of p-MTJ spin-valve is very poor. Thus, this TEM image could not discuss about the f.c.c. crystallinity of the MgO tunnelling barrier.

Response: Thank you very much for this comment. The previous cross-sectional TEM image was not clear enough. We have performed the Cs-corrected TEM to improve the quality and reprofile the crystallisation situation. As shown in Fig. 5a, the excellent crystalline quality of the MgO layers, particularly the lower one acting as the tunnel barrier, should benefit the TMR as one of the factors.

In conclusion, authors did not show any new scientific discovery or any advanced technology in p-MTJ spin-valves.

Response: Thank you. We appreciate the instructive suggestions from the reviewer. The previous studies focused on the W layer-based p-MTJ films provided TMR, thermal stability and other meaningful information. Accordingly, we have optimised the structure and the property of this kind of p-MTJ films using atom-thick W capping

and bridging layers. For the first time, we have demonstrated STT switching by nanopillar fabrication and provided an insight to J_{C0} . The p-MTJ that we have made present a large TMR of 249% and an RA as low as $7 \Omega \cdot \mu\text{m}^2$, which is significant compared to the previous work. Furthermore, we have performed the first-principles calculation to explain the advantages of the atom-thick W layers and attributed that to the resonant tunnelling transmission. We hope this manuscript has become more complete after the revision.

References

1. Song, Y. J. et al. Highly functional and reliable 8Mb STT-MRAM embedded in 28nm logic. In *Electron Devices Meeting (IEDM)*, 2016 IEEE International. 27-2, (2016).
2. Liu, T., Cai, J. W., & Sun, L. Large enhanced perpendicular magnetic anisotropy in CoFeB/MgO system with the typical Ta buffer replaced by an Hf layer. *AIP Adv.* **2**, 032151 (2012).
3. Liu, T., Zhang, Y., Cai, J. W., & Pan, H. Y. Thermally robust Mo/CoFeB/MgO trilayers with strong perpendicular magnetic anisotropy. *Sci. Rep.* **4**, 5895 (2014).
4. An, G. G. et al. Highly stable perpendicular magnetic anisotropies of CoFeB/MgO frames employing W buffer and capping layers. *Acta Mater.* **87**, 259-265 (2015).
5. Peng, S. Z. et al. Origin of interfacial perpendicular magnetic anisotropy in MgO/CoFe/metallic capping layer structures. *Sci. Rep.* **2**, (2015).
6. Peng, S. Z. et al. Giant interfacial perpendicular magnetic anisotropy in MgO/CoFe/capping layer structures. *Appl. Phys. Lett.* **110**, 072403 (2017).
7. Zhou, J. Q. et al. Large influence of capping layers on tunnel magnetoresistance in magnetic tunnel junctions. *Appl. Phys. Lett.* **109**, 242403 (2016).
8. Zhang, B. Y. et al. Influence of heavy metal materials on magnetic properties of Pt/Co/heavy metal tri-layered structures. *Appl. Phys. Lett.* **110**, 012405 (2017).
9. Sabino, M. P. R., Ter Lim, S., & Tran, M. Influence of Ta insertions on the magnetic properties of MgO/CoFeB/MgO films probed by ferromagnetic resonance. *Appl. Phys. Express* **7**, 093002 (2014).
10. Celinski, Z., & Heinrich, B. Exchange coupling in Fe/Cu, Pd, Ag, Au/Fe trilayers. *J. Magn. Mater.* **99**, L25-L30 (1991).
11. Ikeda, S. et al. Tunnel magnetoresistance of 604% at 300 K by suppression of Ta diffusion in CoFeB/MgO/CoFeB pseudo-spin-valves annealed at high temperature. *Appl. Phys. Lett.* **93**, 082508 (2008).
12. Ikeda, S. et al. A perpendicular-anisotropy CoFeB/MgO magnetic tunnel junction. *Nat. Mater.* **9**, 721-724 (2010).

13. Sinha, J. *et al.* Influence of boron diffusion on the perpendicular magnetic anisotropy in Ta/CoFeB/MgO ultrathin films. *J. Appl. Phys.* **117**, 043913 (2015).
14. Chatterjee, J. *et al.* Enhanced annealing stability and perpendicular magnetic anisotropy in perpendicular magnetic tunnel junctions using W layer. *Appl. Phys. Lett.* **110**, 202401 (2017).
15. Kim, J. H. *et al.* Ultrathin W space layer-enabled thermal stability enhancement in a perpendicular MgO/CoFeB/W/CoFeB/MgO recording frame. *Sci. Rep.* **5**, (2015).
16. Kawahara, T., Ito, K., Takemura, R., & Ohno, H. Spin-transfer torque RAM technology: Review and prospect. *Microelectron. Reliab.* **52**, 613-627 (2012).

Reviewer #3 (Remarks to the Author):

In this paper, the authors studied the STT of a p-MTJ with an atomic layer of Tungsten experimentally. It was claimed that the device has large TMR of 249% and at the same time a resistance area product lower than 7.5 ohm micron². First principles study has identified physical origin of high TMR. This is a high-quality work that is suitable for nature comm. However, I have the following concerns that must be clarified.

Response: Thank you very much for noting our work as high-quality. We agree that this work offers a better understanding for the role of the W layers in the TMR enhancement in addition to the large TMR and the low RA that we have obtained. Your suggestions were very instructive and have deeply inspired us. We appreciate it, and hope that this manuscript has become much improved after the revisions.

The punch point of this paper is the achievement of large TMR of 249% and low resistance area product (denoted as RA) lower than 7.5. In Fig.2, results of two devices (the same radius) have been given in Fig.2a and Fig.2c. In Fig.2a, the TMR is 161% while RA is about 7 (see first paragraph on page 5). In Fig.2c, the TMR is 249% while RA is not specified.

Here is my question, looking at Fig.2, the resistances are around R=720 and 950 for Fig.2a and Fig.2c, respectively. Since two devices have the same radius, the RA of Fig.2c is clearly larger than 9 or even close to 10. I would like to know how RA was calculated for Fig.2a, $RA=720 \cdot \pi \cdot r^2$?

To conclude, the statement “TMR of 249% and an RA as low as 7 $\Omega \cdot \mu\text{m}^2$ are simultaneously achieved” is inaccurate. This makes this work not as significant (see Fig.1 of supplementary material).

Response: Thank you. As you have mentioned, the definition of RA included two parts: RA in parallel and anti-parallel states (RA_P and RA_{AP}).^{1,2} For the convenience of comparison, RA generally refers to RA_P .³ In Fig. 2, the resistance in the parallel state was approximately 277 Ω . Thus, RA can be calculated as: $RA = 277 \Omega \times \pi \times 90 \text{ nm}^2 = 7.05 \Omega \cdot \mu\text{m}^2$.

Here are other questions:

(1). In the measurement what is the bias applied?

Response: Thank you. The bias voltage we applied in the STT measurement was from $\pm 0.5 \text{ V}$ to $\pm 0.95 \text{ V}$. The specific bias voltage we set for each measurement was

determined by the largest resistance of the p-MTJ nanopillar. For example, in the case of Fig. 2b, the bias voltage was approximately 0.68 V.

(2). Caption for Fig.2c and Fig.2d are missing.

Response: Thank you. We have corrected this mistake in the revised manuscript.

(3). The resonant tunnelling was identified as the origin of high TMR. Since resonant tunnelling is vulnerable to bias and impurity, this issue has to be addressed.

Response: Thank you. We have addressed this concern by further calculating the dependence on the bias voltage (V_b) of TMR using the atomic structures we built in the main text. The TMR for the p-MTJ with the W insertion dramatically dropped with the increasing V_b : the TMR was 173% when $V_b = 10$ mV and decreased to 107% when $V_b = 50$ mV. In contrast, the TMR for the p-MTJ with the Ta insertion remained to be 88% at $V_b = 50$ mV. This intense TMR decay can also be observed in the experimental measurement shown in Figs. 2b and d and Fig. 3b in the manuscript. The p-MTJ using Ta insertion presented less dependency.⁴ Therefore, we believe that the atom-thick W insertion-induced resonant tunnelling transmission indeed contributed to the TMR enhancement.

(4). Fig.2c, in the inset it says TMR=248% while TMR=249% in the text.

Response: Thank you. We apologise for that. We have corrected this mistake accordingly.

(5). Since the presence of W layer is detrimental to resonant tunnelling (make the system asymmetric), what is the role play by W, to help stabilize the system? Or to help the system to crystalize?

Response: Thank you. In addition to the resonant tunnelling transmission, the advantages of the atom-thick W layers can be summarised as follows:

First, as the reviewer has mentioned, the W insertions allowed for an excellent crystalline quality of the MgO barrier. This effect has been verified from the Cs-corrected TEM (Fig. 5a) that we performed in the revised manuscript. The W insertions enabled robust interfacial PMA compared to Ta and other conventional materials.⁵ Moreover, the EELS results in Fig. 5b and Supplementary Fig. 8 showed that the diffusion of the W atoms at 410 °C annealing was suppressed in contrast with Ta. The W insertions acted as B absorbers. The strong resistance to the higher

temperature further improved the MgO and CoFeB crystallinity, thereby resulting in a larger TMR.

Second, the J_C we obtained was comparable to that for the p-MTJs configured with the Ta layers and did not scale with the enhanced Δ , which can be ascribed to the lower damping constant (α). The α of p-MTJs using the Ta layers was sub-optimal for the strong spin-orbit coupling and atom diffusion at the high temperature. Our earlier experiments proved that α was material dependent, and, thus, could be reduced by replacing Ta with W as the capping layer.⁶

References

1. Parkin, S. S. P. *et al.* Giant tunnelling magnetoresistance at room temperature with MgO (100) tunnel barriers. *Nat. Mater.* **3**, 862-867 (2004).
2. Yuasa, S., Nagahama, T., Fukushima, A., Suzuki, Y., & Ando, K. Giant room-temperature magnetoresistance in single-crystal Fe/MgO/Fe magnetic tunnel junctions. *Nat. Mater.* **3**, 868-871 (2004).
3. Ikeda, S. *et al.* A perpendicular-anisotropy CoFeB/MgO magnetic tunnel junction. *Nat. Mater.* **9**, 721-724 (2010).
4. Devolder, T., Le Goff, A., & Nikitin, V. Size dependence of nanosecond-scale spin-torque switching in perpendicularly magnetized tunnel junctions. *Phys. Rev. B* **93**, 224432 (2016).
5. An, G. G. *et al.* Highly stable perpendicular magnetic anisotropies of CoFeB/MgO frames employing W buffer and capping layers. *Acta Mater.* **87**, 259-265 (2015).
6. Zhang, B. Y. *et al.* Influence of heavy metal materials on magnetic properties of Pt/Co/heavy metal tri-layered structures. *Appl. Phys. Lett.* **110**, 012405 (2017).

Reviewers' comments:

Reviewer #1 (Remarks to the Author):

In the revised version, the authors have performed first principles calculations further to address the critical comments. The calculated results, including the TMR at finite bias voltages, are likely to be consistent with the experimental observations. However, it does not appear that the results added are sufficient to attribute the experimental observations to the mechanism proposed theoretically. So, I am still wondering which causes the large TMR, “the electronic effect of the atomic W layer” or “the ingenious optimization in the sample preparation process”.

If the authors are aiming at clarifying the electronic effect of the atomic W layer on TMR, it would be better to examine not only the bias voltage dependence but also the W layer number dependence of TMR. In the previous studies, it was shown that even thicker W layers are effective to enhance the TMR (for example, TMR = 248%, RA = ~10 Ohm.micro-m²: Tezuka et al., IEEE Magn. Lett. 7, 3104204 (2016)).

Reviewer #2 (Remarks to the Author):

I reviewed the author's answer about the issues and questions. Although the author showed very high TMR ratio of p-MTJ spin-valve, the quality of experimental data should be improved and also the mechanism why the p-MTJ spin-valve achieved high TMR ratio should be evidently explained.

There are some issues that authors should explain evidently:

1. Authors explained that “Fig. 5a shows a Cs-corrected TEM image of the p-MTJ stack annealed at 390 °C, which verifies the excellent crystalline quality of the MgO barrier”. However, the TEM image of Fig.5a could not observe the f.c.c. crystallinity of the MgO barrier layer. In addition, the EELS intensity could not say “no distribution change of W is shown compared with the nominal locations of W spacer and bridging layers at increasing temperatures”, since the EELS could not clearly distinguish the W distribution because of low radial-distribution resolution. It is strongly recommended that the author should improve the quality of the experimental data. (i.e. high resolution x-TEM observation of the CoFeB/MgO interface region1 or compositional depth profile by high resolution SIMS)
2. Authors mentioned a higher TMR ratio of p-MTJ using W spacer was associated with the resonance tunnelling effect of the W spacer in Fig. 1a. If it is true, the TMR ratio should oscillate when the W spacer thickness varies, as shown in Fig. 1a. Authors should prove that the TMR ratio oscillates when the W spacer thickness is changed(ref 1,2). If the TMR ratio peaks at a

specific W thickness, it would be related to the inter-diffusion between the upper and lower CoFeB layers at a thin W spacer thickness or the non-ferro-coupling between upper and lower CoFeB layers³. It is strongly recommended that the author add the dependency of W spacer thickness on TMR ratio.

3. In general, people do an ex-situ annealing with a high perpendicular magnetic field (i.e., 3 Tesla) on p-MTJ spin-valve before the CIPT measurement, which saturates the electron spin direction of the perpendicular magnetic free layer. In this manuscript, there is no indication of the applied magnetic field during an ex-situ annealing. If author did an ex-situ annealing with a high perpendicular magnetic field prior to the CIPT measurement, the TMR ratio should not be changed after applying the perpendicular magnetic field of 2 kOe shown in figure 2. Author should explain the reason why the TMR ratio increased after applying the perpendicular magnetic field of 2 kOe.

4. In the reply for my comment 2, author wrote “Further, the thinner the W spacer layer is, the lower the α becomes, because W diffusion is weakened with decreasing thickness.⁹” The reference 9 had no relation supporting your reply. The paper explored the influence of Ta insertion by sputtering additional layers of Ta/CoFeB. The thickness of the tantalum was fixed at 0.3 nm. The inter-diffusion of the Ta seems to be greater because of the inter-mixing from multiple interfaces and not the thick thickness of a spacer layer.

References

1. A. N. Useinov* and J. Kosel, Resonant tunnel magnetoresistance in double-barrier planar magnetic tunnel junctions. PHYSICAL REVIEW B 84, 085424 (2011)
2. C. H. Chen and W. J. Hsueh. Enhancement of tunnel magnetoresistance in magnetic tunnel junction by a superlattice barrier. Appl. Phys. Lett. 104, 042405 (2014).
3. D. Y. Lee, et al. Dependency of Tunneling-Magnetoresistance Ratio on Nanoscale Spacer Thickness and Material for Double MgO Based Perpendicular-Magnetic-Tunneling-Junction. Sci. Rep. 6, 38125 (2016).

Reviewer #3 (Remarks to the Author):

In the revision, the authors answered all my questions in a satisfactory way. I think the present version is ready for publication.

Reviewer #1 (Remarks to the Author):

In the revised version, the authors have performed first principles calculations further to address the critical comments. The calculated results, including the TMR at finite bias voltages, are likely to be consistent with the experimental observations. However, it does not appear that the results added are sufficient to attribute the experimental observations to the mechanism proposed theoretically. So, I am still wondering which causes the large TMR, “the electronic effect of the atomic W layer” or “the ingenious optimization in the sample preparation process”.

Response: Thank you again for the approval of our work regarding STT switching in atom-thick engineered p-MTJs with large TMR and low RA. We appreciate the instructive comments, which helped us further enhance the quality of the manuscript. We hope that this manuscript has become much improved after the 2nd revision.

If the authors are aiming at clarifying the electronic effect of the atomic W layer on TMR, it would be better to examine not only the bias voltage dependence but also the W layer number dependence of TMR. In the previous studies, it was shown that even thicker W layers are effective to enhance the TMR (for example, TMR = 248%, RA = ~10 Ohm.micro-m2: Tezuka et al., *IEEE Magn. Lett.* 7, 3104204 (2016)).

Response: Thank you very much for this suggestion, which helped us present more clearly the relationship among TMR, film thickness, magnetic coupling, and STT switching performance for our devices.

First, we investigate TMR in terms of the W spacer layer thickness by performing a first-principles calculation. The atomic structures were built according to our experimental p-MTJ configuration, including 7 CoFe/5 MgO/9 CoFe/1, 3 W/5 CoFe (see Supplementary Fig. 6, numbers denote the amount of atom layers). As shown in Table 1, the calculated TMR decreases from 245% to 162% while increasing the thickness of W spacer layer from one single atom to three (0.58 nm).

Table 1 | Spin-resolved conductance and TMR values in CoFe/MgO/CoFe/W (Ta)/5 CoFe structures with various film thicknesses. The conductance unit is e^2/h .

MTJ film	Conductance (P)	Conductance (AP)	TMR
9 CoFe/1 W/5 CoFe	5.59×10^{-5}	1.62×10^{-5}	245%
9 CoFe/1 Ta/5 CoFe	3.92×10^{-5}	2.07×10^{-5}	89%
9 CoFe/3 W/5 CoFe	3.13×10^{-5}	1.20×10^{-5}	162%
11 CoFe/3 W/7 CoFe	6.10×10^{-5}	1.36×10^{-5}	348%

Then, we modelled the p-MTJ stack with thicker upper and bottom CoFeB free layers, *i.e.*, 7 CoFe/5 MgO/11 CoFe/3 W/7 CoFe. It revealed that for the majority spin

in P state, the spin-resolved conductance increased with enhanced coherent tunnelling. A TMR as high as 348% can be obtained by using thicker CoFeB free layers, thus the large TMR of 248% reported by Tezuka N. *et al.* can be explained.¹ It is important to mention that the TMR depends on the two CoFeB free layers, W spacer layer, and MgO barriers, making it difficult to determine the configuration possessing the largest TMR. However, our main objective is to realize ultrahigh TMR with STT switching, which was not shown in Ref. 1 (three-atom W layers), thereby different device structure was investigated in our experiments to achieve our objective. On the other hand, we experimentally studied the situation of anisotropy field (H_k) in terms of CoFeB thicknesses (t), as shown in Fig. 1a (Fig. 1d in the main text). For the p-MTJ film with $t = 1.3$ nm, H_k is around 3102 Oe, allowing a compromise between high thermal stability and low threshold current density.

In order to obtain STT switching in the p-MTJs with a double MgO/CoFeB interface free layer, the two CoFeB free layers have to reverse simultaneously by ferromagnetic coupling.² For that purpose, we performed the first-principles calculation to study how the thickness of W spacer layer influences the magnetic coupling between the two CoFeB free layers. As plotted in Fig. 1b, the two CoFe free layers present a strong ferromagnetic coupling while using single-atom W spacer layer, while a weak antiferromagnetic coupling with three-atom W layers. As also shown in the experiment reported in Ref. 3, the two CoFeB layers coupled antiferromagnetically via a 0.7 nm spacer layer, which is quite negative for the STT mechanism. While in our experiment, the 0.2-nm W spacer layer enables the two CoFeB free layers to switch as a single layer in STT measurement. Besides, W is a heavy metal with strong spin orbit coupling. By using a thinner W spacer layer, the spin polarization can be retained from being scattered while transporting between two CoFeB free layers. Therefore, the atom-thick W spacer layer is quite critical for the switching reliability of our device regarding STT performance.

Figure 1 | (a) Dependence of H_k on t . (b) Magnetic coupling between two CoFeB free layers changes with increasing thickness of the W spacer layer.

References

1. Tezuka, N. *et al.* Perpendicular magnetic tunnel junctions with low resistance-area product: high output voltage and bias dependence of magnetoresistance. *IEEE Magn. Lett.* **7**, 3104204 (2016).
2. Sato, H., Yamanouchi, M., Ikeda, S., Fukami, S., Matsukura, F., & Ohno, H. Perpendicular-anisotropy CoFeB-MgO magnetic tunnel junctions with a MgO/CoFeB/Ta/CoFeB/MgO recording structure. *Appl. Phys. Lett.* **10**, 022414 (2012).
3. Lee, D. Y., Hong, S. H., Lee, S. E., & Park, J. G. Dependency of tunnelling-magnetoresistance ratio on nanoscale spacer thickness and material for double MgO based perpendicular-magnetic-tunnelling-junction. *Sci. Rep.* **6**, 38125 (2016).

Reviewer #2 (Remarks to the Author):

I reviewed the author's answer about the issues and questions. Although the author showed very high TMR ratio of p-MTJ spin-valve, the quality of experimental data should be improved and also the mechanism why the p-MTJ spin-valve achieved high TMR ratio should be evidently explained.

Response: We would like to thank you again for the comments and suggestions to improve the quality of our manuscript. We have taken all the comments seriously and revised our manuscript accordingly. We hope our work is presented more clearly after the 2nd revision.

There are some issues that authors should explain evidently:

1. Authors explained that “Fig. 5a shows a Cs-corrected TEM image of the p-MTJ stack annealed at 390 °C, which verifies the excellent crystalline quality of the MgO barrier”. However, the TEM image of Fig.5a could not observe the f.c.c. crystallinity of the MgO barrier layer. In addition, the EELS intensity could not say “no distribution change of W is shown compared with the nominal locations of W spacer and bridging layers at increasing temperatures”, since the EELS could not clearly distinguish the W distribution because of low radial-distribution resolution. It is strongly recommended that the author should improve the quality of the experimental data. (i.e. high-resolution x-TEM observation of the CoFeB/MgO interface region1 or compositional depth profile by high resolution SIMS)

Response: Thank you very much for these suggestions. We performed Cs-corrected TEM again in the revision. As shown in Fig. 1, Fig. 1a was performed with high-resolution TEM, which we first submitted; Fig. 1a was obtained by Cs-corrected TEM (model JEM ARM 200F), of which the resolution is 0.08 nm. Having analysed all of the previous TEM images, we believe that the Cs-corrected TEM is the most preferred tool regarding our p-MTJ films, and the crystallinity of MgO barrier can be seen clearly from Fig. 1b.

The EELS can provide at least the same or even higher resolution as compositional depth profile by high-resolution SIMS,¹⁻³ and the equipment JEOL-2100F we used in this work has a resolution of 0.2 nm. While considering W is a heavy metal element, we additionally carried out an EDS test this time. As mapped in Fig. 1c, no signal of W was detected within the MgO barriers or at the CoFeB/MgO interfaces, supporting the point that W is robust against high temperature diffusion. As one of the origins for large TMR, the robustness of W insertion, in addition to the crystallinity of MgO barrier, is underlined in our work.

Figure 1 | Cross-sectional (a) high-resolution TEM and (b) Cs-corrected TEM images of the p-MTJ stacks. (c) EDS mapping of the p-MTJ stack, where W is in red.

2. Authors mentioned a higher TMR ratio of p-MTJ using W spacer was associated with the resonance tunnelling effect of the W spacer in Fig. 1a. If it is true, the TMR ratio should oscillate when the W spacer thickness varies, as shown in Fig. 1a. Authors should prove that the TMR ratio oscillates when the W spacer thickness is changed (ref 1,2). If the TMR ratio peaks at a specific W thickness, it would be related to the inter-diffusion between the upper and lower CoFeB layers at a thin W spacer thickness or the non-ferro-coupling between upper and lower CoFeB layers.³ It is strongly recommended that the author add the dependency of W spacer thickness on TMR ratio.

Response: Thank you very much for this suggestion. In order to obtain STT switching in the p-MTJs with a double MgO/CoFeB interface free layer, the two CoFeB free layers have to reverse simultaneously by ferromagnetic coupling.⁴ For that purpose, we performed a first-principles calculation to study how the thickness of W spacer layer influences the magnetic coupling between the two CoFeB free layers first. As plotted in Fig. 2, the two CoFe free layers present a strong ferromagnetic coupling while using single-atom W spacer layer, while a weak antiferromagnetic coupling with three-atom W layers. As shown in the experiment reported in Ref. 5, the two CoFeB layers coupled antiferromagnetically via a 0.7 nm spacer layer, which is quite negative for the STT mechanism. While in our experiment, the 0.2-nm W spacer layer enables the two CoFeB free layers to switch as a single layer in STT measurement.

Figure 2 | Magnetic coupling between two CoFeB free layers changes with increasing thickness of the W spacer layer.

From the view point of STT switching, the calculation of TMR in the case of ferromagnetic coupling is more meaningful. We studied the dependence of TMR on W insertion thickness by the first-principles calculation, where the calculated TMR decreases from 245% to 162% while increasing the thickness of W from one-atom to three-atom layer. It illustrates that the TMR varies with the thickness of W insertion, and single-atom layer W shows larger TMR. This tendency is consistent with our experimental results. Even-number (e.g., two) atom layer W was not considered, as the relative positions of Co atom and W atom would be changed from top-hollow site to top-top site at the CoFe/W interface, which will involve additional parameters for the TMR comparison. In order to avoid the confusion, only odd-number atom layer W were taken into account, this method is common for the theoretical investigation of TMR study in MgO/CoFe multilayers, as shown in Ref. 6. Five and more atom layers are too thick to provide ferromagnetic coupling and STT mechanism. Thereby, only two W thicknesses MTJ stacks have been studied.

Additionally, it is worth noting that the resonant tunnelling transmission is sensitive to the bias voltage (V_b). Thus, we calculated the dependence on the V_b of TMR using the atomic structures we built in the main text. For the p-MTJ with the single-atom W insertion, the TMR dramatically dropped with the increasing V_b : the TMR was 173% when $V_b = 10$ mV and decreased to 107% when $V_b = 50$ mV. On the contrary, the TMR for the p-MTJ with the Ta insertion remained 88% at $V_b = 50$ mV. This intense TMR decay associated with W insertion was also observed in Figs. 2b and 3b in the main text. The p-MTJ using the Ta insertion presented less dependency.⁷ Therefore, we think that the W insertion-induced resonant tunnelling transmission could contribute to the TMR enhancement. Moreover, considering the complexity of the p-MTJ films, we do not exclude the possibility that there are other contributions coexisting in this system.

3. In general, people do ex-situ annealing with a high perpendicular magnetic field

(i.e., 3 Tesla) on p-MTJ spin-valve before the CIPT measurement, which saturates the electron spin direction of the perpendicular magnetic free layer. In this manuscript, there is no indication of the applied magnetic field during an ex-situ annealing. If author did ex-situ annealing with a high perpendicular magnetic field prior to the CIPT measurement, the TMR ratio should not be changed after applying the perpendicular magnetic field of 2 kOe shown in figure 2. Author should explain the reason why the TMR ratio increased after applying the perpendicular magnetic field of 2 kOe.

Response: We apologise for the unclear statement in the manuscript. We did the ex-situ annealing without magnetic field and the electron spin direction of the free layer is not saturated during the CIPT measurement. To avoid the confusion, we have removed the intermediate results regarding the change of TMR.

4. In the reply for my comment 2, author wrote “Further, the thinner the W spacer layer is, the lower the α becomes, because W diffusion is weakened with decreasing thickness.”⁹ The reference 9 had no relation supporting your reply. The paper explored the influence of Ta insertion by sputtering additional layers of Ta/CoFeB. The thickness of the tantalum was fixed at 0.3 nm. The inter-diffusion of the Ta seems to be greater because of the inter-mixing from multiple interfaces and not the thick thickness of a spacer layer.

Response: Thank you very much for the comment. We cite Ref. 3 to illustrate our point upon the atom diffusion with respect to the film thickness. For the structure of the Ta underlayer/CoFeB/MgO/Ta, volume-averaged magnetization (M/V) decreases monotonically with increasing thickness of Ta underlayer. The reduction of M/V indicates the formation of an additional magnetic dead layer. In addition, the magnetic dead layer is formed when diffusion of Ta occurs through the Ta/CoFeB interface. Combining this piece of information, we concluded that “the thinner the W spacer layer is, the lower the α becomes, because W diffusion is weakened with decreasing thickness”.

References

1. Wang, Z. *et al.* Atomic-scale structure and local chemistry of CoFeB-MgO magnetic tunnel junctions. *Nano Lett.* **16**, 1530 (2016).
2. Miyajima, T. *et al.* Transmission electron microscopy study on the crystallization and boron distribution of CoFeB/MgO/CoFeB magnetic tunnel junctions with various capping layers. *Appl. Phys. Lett.* **94**, 122501 (2009).
3. Sinha, J. *et al.* Influence of boron diffusion on the perpendicular magnetic anisotropy in Ta/CoFeB/MgO ultrathin films. *J. Appl. Phys.* **117**, 043913 (2015).
4. Sato, H., Yamanouchi, M., Ikeda, S., Fukami, S., Matsukura, F., & Ohno, H. Perpendicular-anisotropy CoFeB-MgO magnetic tunnel junctions with a

- MgO/CoFeB/Ta/CoFeB/MgO recording structure. *Appl. Phys. Lett.* **10**, 022414 (2012).
5. Lee, D. Y., Hong, S. H., Lee, S. E., & Park, J. G. Dependency of tunnelling-magnetoresistance ratio on nanoscale spacer thickness and material for double MgO based perpendicular-magnetic-tunnelling-junction. *Sci. Rep.* **6**, 38125 (2016).
 6. Sankaran, K., Swerts, J., Couet, S., Stokbro, K., & Pourtois, G. Oscillatory behavior of the tunnel magnetoresistance due to thickness variations in Ta/CoFe/MgO magnetic tunnel junctions: A first-principles study. *Phys. Rev. B* **94**, 094424 (2016).
 7. Devolder, T., Le Goff, A., & Nikitin, V. Size dependence of nanosecond-scale spin-torque switching in perpendicularly magnetized tunnel junctions. *Phys. Rev. B* **93**, 224432 (2016).

Reviewer #3 (Remarks to the Author):

In the revision, the authors answered all my questions in a satisfactory way. I think the present version is ready for publication.

Response: Thank you for the support for our study regarding STT switching in atom-thick engineered p-MTJs with large TMR and low RA . We really appreciate your instructive comments and suggestions, which have led us to demonstrate our work more completely and systematically. Thank your again for your decision of acceptance, we will launch more in-depth research to pave the way to close the gap for STT-MRAM and other intriguing topics in spintronics.

Reviewers' comments:

Reviewer #1 (Remarks to the Author):

The authors addressed the W-thickness dependence of TMR through first principles calculations, as I suggested. The calculation results and their comparison with the experiments appear to be useful information to discuss the mechanism of the enhanced TMR. Thus, I personally think the manuscript that includes the responses to the reviewer's comments would be acceptable for publication, although the TMR enhancement mechanism has not really been clarified yet. In short, I would like to strongly recommend that the authors should make the following revisions finally:

- (1) Table 1 in the authors' response, i.e., the W thickness dependence of calculated TMR etc., should be shown in Supplementary Information.
- (2) As the authors said, the related results by Tezuka et al. are also explained by the present calculations. Therefore, "Tezuka et al. IEEE Magn. Lett. (2016)" should be listed as a reference, as well as the paper of "Lee et al. NPG Asia (2016)." In addition, the data points of these two papers should be added in Supplementary Figure 1 (TMR vs. RA for p-MTJs). This makes the present paper fair and reliable.

Reviewer #2 (Remarks to the Author):

I have gone through manuscript again, but not yet convinced of which is the dominant factor contributing to the high TMR ratio.

1. The quality of the tunneling barrier of p-MTJ using Ta or W,
2. The resonance tunneling of the p-MTJ using tungsten)

I hope the following questions can help clarify the issue.

1. In my previous question 3, author answered that CIPT sample was measured before the saturation of the electron spin direction. In other works [1], CIPT measurements show correlation with the TMR measured from R-H measurement. If the author claims that the high TMR ratio of 249% was obtained, the CIPT measurement data should be consistent and include the TMR ratio after saturation of the electron spin direction to compare with the data of the R-H curve. Considering the fact that the p-MTJ is likely to be damaged during patterning of the p-MTJ, the CIPT measurement should be similar to the value calculated from the R-H curve or higher. It would also improve the work if the author can provide TMR ratio of the p-MTJs with respect to the junction size as in reference 2.

2. As I mentioned above the origin of the high TMR ratio still seems unclear. Although the first-

principle calculation shows that the tunnelling transmission is higher for the p-MTJ using tungsten compared to tantalum, the crystallinity issue still remains. In the SIMS data (figure 4) of reference [3], it showed that lower TMR ratio of p-MTJ using tantalum is due to the Ta diffusion into the MgO degrading the crystallinity of tunnelling barrier. It would strengthen the author's claim if the quality of the MgO tunnel barrier of the p-MTJ using tantalum is similar to that of p-MTJ using tungsten.

References

1. Y. J. Song et al., "Highly functional and reliable 8Mb STT-MRAM embedded in 28nm logic," 2016 IEEE International Electron Devices Meeting (IEDM), San Francisco, CA, 2016, pp. 27.2.1-27.2.4.
2. S. Ikeda et al., "Perpendicular-anisotropy CoFeB-MgO based magnetic tunnel junctions scaling down to 1X nm," 2014 IEEE International Electron Devices Meeting, San Francisco, CA, 2014, pp. 33.2.1-33.2.4.
3. Lee, D Y., Hong, S. H., Lee, S. E., & Park, J. G. Dependency of tunnelling-magnetoresistance ratio on nanoscale spacer thickness and material for double MgO based perpendicular-magnetic-tunnelling- junction. *Sci. Rep.* 6, 38125 (2016).

Response to referees letter

First of all, we gratefully thank the two reviewers for the time and efforts spent on reviewing our manuscript and providing us very detailed review comments. We found all of their comments constructive and useful, and we have taken them into account and carefully revised the paper. The revisions have significantly improved the paper, and we hope the manuscript now meets the requirements for publication. The major changes in the manuscript are highlighted in blue. Below we provide our point-to-point responses to all of the review comments.

Reviewer #1 (Remarks to the Author):

The authors addressed the W-thickness dependence of TMR through first principles calculations, as I suggested. The calculation results and their comparison with the experiments appear to be useful information to discuss the mechanism of the enhanced TMR. Thus, I personally think the manuscript that includes the responses to the reviewer's comments would be acceptable for publication, although the TMR enhancement mechanism has not really been clarified yet.

Response from Authors: Thank you so much for evaluating our work as “acceptable after revision”. We really appreciate your support for our study regarding STT switching in atom-thick engineered p-MTJs with large TMR and low RA. Thanks to your instructive comments and suggestions, our work has been demonstrated more completely and systematically.

The TMR enhancement mechanism can be explained briefly by two factors:

1. The better quality of MgO barrier for p-MTJs using W spacer layer thanks to the higher annealing temperature. This is proved by the Cs-corrected TEM, EELS and EDS experiments.
2. The resonant tunnelling of the p-MTJs using W thanks to the higher density of scattering state. This is proved by the systematic first-principles calculation and the experimental TMR dependence on the W layer thickness.

We have performed revision in the main text to present these points more clearly.

In short, I would like to strongly recommend that the authors should make the following revisions finally:

- (1) Table 1 in the authors' response, i.e., the W thickness dependence of calculated TMR etc., should be shown in Supplementary Information.

Response from Authors: Thank you very much for the comment. We have added the table you mentioned to Supplementary Information as Supplementary Note 9, which is stated as below:

We further established the first-principles calculations to present the relationship between TMR and film thickness of our devices more clearly.

First, we investigate TMR in terms of the W spacer layer thickness. The atomic structures were built according to our experimental p-MTJ configuration, including 7 CoFe/5 MgO/9 CoFe/1, 3 W/5 CoFe. As shown in Supplementary Table 1, the calculated TMR decreases from 245% to 162% while increasing the thickness of W spacer layer from one single atom to three (0.58 nm). Then, we modelled the p-MTJ stack with thicker upper and bottom CoFeB free layers, i.e., 7 CoFe/5 MgO/11 CoFe/3 W/7 CoFe. It revealed that for the majority spin in P state, the spin-resolved conductance increased with enhanced coherent tunnelling. A TMR as high as 348% can be obtained by using thicker CoFeB free layers. Thus, the TMR depends on the two CoFeB free layers, W spacer layer, and MgO barriers, making it difficult to determine the configuration possessing the largest TMR. However, our main objective is to realize ultrahigh TMR with STT switching, thereby different device structure was investigated in our experiments to achieve this phenomenon.

Supplementary Table 1 | Spin-resolved conductance and TMR values in 7 CoFe/5 MgO/CoFe/W(Ta)/CoFe structures with various film thicknesses. The conductance unit is e^2/h .

MTJ film	Conductance (P)	Conductance (AP)	TMR
9 CoFe/1 W/5 CoFe	5.59×10^{-5}	1.62×10^{-5}	245%
9 CoFe/1 Ta/5 CoFe	3.92×10^{-5}	2.07×10^{-5}	89%
9 CoFe/3 W/5 CoFe	3.13×10^{-5}	1.20×10^{-5}	162%
11 CoFe/3 W/7 CoFe	6.10×10^{-5}	1.36×10^{-5}	348%

(2) As the authors said, the related results by Tezuka et al. are also explained by the present calculations. Therefore, “Tezuka et al. IEEE Magn. Lett. (2016)” should be listed as a reference, as well as the paper of “Lee et al. NPG Asia (2016).” In addition, the data points of these two papers should be added in Supplementary Figure 1 (TMR vs. RA for p-MTJs). This makes the present paper fair and reliable.

Response from Authors: Thank you very much for the comment. The paper “Lee et al. NPG Asia (2016)” is Ref. 35 in the main text, and we have additionally cited “Tezuka et al. IEEE Magn. Lett. (2016)” as Ref. 36 in the revision. However, Supplementary Figure 1 focuses on data that have been published with STT switching. Considering these two works did not demonstrate STT switching, we think it may be fair to cite them literally other than in the picture.

We have performed revision in the Supplementary Note 1 to annotate supplementary figure 1 more clearly. The relevant sentence has been rewritten as: “It

should be mentioned that though large TMRs have been found in the p-MTJ films similar to ours, no nano-pillar devices or STT switching have been presented currently for our collection.¹⁻²”

References

1. Lee, S. E., Shim, T. H., & Park, J. G. Perpendicular magnetic tunnel junction (p-MTJ) spin-valves designed with a top Co₂Fe₆B₂ free layer and a nanoscale-thick tungsten bridging and capping layer. *NPG Asia Mater.* **9**, e324 (2016).
2. Tezuka, N. et al. Perpendicular magnetic tunnel junctions with low resistance-area product: high output voltage and bias dependence of magnetoresistance. *IEEE Magn. Lett.* **7**, 3104204 (2016).

Reviewer #2 (Remarks to the Author):

I have gone through manuscript again, but not yet convinced of which is the dominant factor contributing to the high TMR ratio.

1. The quality of the tunnelling barrier of p-MTJ using Ta or W,
2. The resonance tunnelling of the p-MTJ using tungsten)

Response from Authors: Thank you again for the comments and suggestions that you have provided, which helped us come to a better explanation of the origins for the large TMR. We hope that this manuscript has become improved after the revision.

The TMR enhancement mechanism can be explained mainly by two factors:

1. The better quality of MgO barrier for p-MTJs using W spacer layer thanks to the higher annealing temperature, which is proved by the Cs-corrected TEM, EELS and EDS experiments. This contribution has also been verified by the pioneering work Ref.1 that you mentioned.
2. The resonant tunnelling of the p-MTJs using W thanks to the higher density of scattering state, which is proved by the systematic first-principles calculation and the experimental TMR dependence on the W layer thickness.

We think that both factors play the role in TMR enhancement and it is difficult to conclude which one is dominant.

We have performed major revision in the discussion part of the main text to present these points more clearly.

I hope the following questions can help clarify the issue.

1. In my previous question 3, author answered that CIPT sample was measured before the saturation of the electron spin direction. In other works [1], CIPT measurements show correlation with the TMR measured from R-H measurement. If the author claims that the high TMR ratio of 249% was obtained, the CIPT measurement data should be consistent and include the TMR ratio after saturation of the electron spin direction to compare with the data of the R-H curve. Considering the fact that the p-MTJ is likely to be damaged during patterning of the p-MTJ, the CIPT measurement should be similar to the value calculated from the R-H curve or higher.

Response from Authors: Thank you very much for the suggestions. The results of CIPT measurement regarding atom-thick W based p-MTJ films have been shown in Supplementary Note 2 and Supplementary Fig. 2a. To be honest, we are unable to perform annealing with perpendicular magnetic field to saturate the electron spin direction, as our p-MTJ films were deposited on 200 mm wafers, and we do not have furnace with large-scale perpendicular magnetic field that can perform vacuum annealing towards full wafers before CIPT. The fabrication flow was also developed with 200 nm wafers. This is an important reason that we prefer to measure the TMR

directly from nanopillars. It should be mentioned that the first reference you cited (Ref. 2 here) also shows the TMR results of device, a maximum TMR more than 215% after full integration is noted and the STT switching curve demonstrates an average TMR around 180%, which is considered as an average value. In fact, for those p-MTJs annealed without magnetic field, the TMR directly measured from nanopillars is usually more reliable than that from CIPT method.³⁻⁴

Interestingly, Ref.4 shows the comparison between CIPT and device TMR value, they used annealing condition similar to ours (400° without magnetic field) and presents a higher TMR ratio after the device fabrication (the average value increases from 140% to 149% and some device TMRs are higher than 170%). This increase of TMR from CIPT to device measurement agrees with the tendency observed in our experiments.

It would also improve the work if the author can provide TMR ratio of the p-MTJs with respect to the junction size as in reference 2.

Response from Authors: Thank you very much for the interesting suggestion. Our primary motivation is to demonstrate STT switching mechanism in atom-thick W engineered p-MTJ nano-pillars with the largest TMR, which has never been reported in the similar structures. Junction size of 100 nm is able to demonstrate STT switching, and it took us a year to develop this nanopillar technology. We understand that you are expecting more advanced achievement from us, whereas this is deviating from our original motivation. The junction size from 100 nm to 1x nm requires new fabrication technology process and we expect to explore this in the future.

2. As I mentioned above the origin of the high TMR ratio still seems unclear. Although the first-principle calculation shows that the tunnelling transmission is higher for the p-MTJ using tungsten compared to tantalum, the crystallinity issue still remains. In the SIMS data (figure 4) of reference [3], it showed that lower TMR ratio of p-MTJ using tantalum is due to the Ta diffusion into the MgO degrading the crystallinity of tunnelling barrier. It would strengthen the author's claim if the quality of the MgO tunnel barrier of the p-MTJ using tantalum is similar to that of p-MTJ using tungsten.

Response from Authors: Thank you very much for this comment. We agree that the suppression of atom diffusion by using W insertion, or the improvement of MgO quality, contributes to the enhancement of TMR. This point has already been highlighted in the abstract “the robustness of W layers against high temperature diffusion avoids TMR degradation during annealing”. The reference that you mentioned (Ref.1 here) clearly showed a better crystallinity of the p-MTJ using W

than that using Ta under the same annealing condition. This pioneering work has given us lots of inspiration and it has already been in our reference list.

We made great efforts to further optimize the result of Cs-corrected TEM in this revision, as you suggested previously. As shown in the Figure below (Fig. 5 in the main text), the good crystallinity of MgO barrier for the p-MTJ using W layers has been clearly illustrated.

Figure | Cs-corrected TEM and EELS results. (a) Cs-corrected TEM image that profiles the crystallization. The p-MTJ stack was annealed at 390 °C. (b) EELS intensities of Mg, B, and W. Arrows show the positions of the same layer in the two figures. (c) EDS mapping of the p-MTJ stack, where W is in red.

References

1. Lee, D. Y., Hong, S. H., Lee, S. E., & Park, J. G. Dependency of Tunnelling-Magnetoresistance Ratio on Nanoscale Spacer Thickness and Material for Double MgO Based Perpendicular-Magnetic-Tunnelling-Junction. *Sci. Rep.* **6**, (2016).
2. Song, Y. J. et al. Highly functional and reliable 8Mb STT-MRAM embedded in 28nm logic. *In Electron Devices Meeting (IEDM)*, 2016 IEEE International. 27-2, (2016).
3. Ikeda, S. et al. A perpendicular-anisotropy CoFeB/MgO magnetic tunnel junction. *Nat. Mater.* **9**, 721–724 (2010).
4. Park, C. et al. Systematic optimization of 1 Gbit perpendicular magnetic tunnel junction arrays for 28 nm embedded STT-MRAM and beyond. *In Electron Devices Meeting (IEDM)*, 2015 IEEE International. 26-2, (2015).

Finally, we hope the revised manuscript can now meet the requirements for publication.

With our kind regards,

Weisheng Zhao and Albert Fert (on behalf of the authors)

Reviewers' Comments:

Reviewer #1 (Remarks to the Author):

I feel that the authors have addressed my comments well.

Reviewer #2 (Remarks to the Author):

The authors addressed all my questions accordingly. Even though the enhanced TMR mechanism is still unclear, I think calculation results and the comparison data will be valuable information unless it is proved otherwise. I think the manuscript is acceptable for publication.